# USP49 potently stabilizes APOBEC3G protein by removing ubiquitin and inhibits HIV-1 replication

Ting Pan[1], Zheng Song[1], Liyang Wu[1], Guangyan Liu[1], Xiancai Ma[1], Zhilin Peng[1], Mo Zhou[1], Liting Liang[1], Bingfeng Liu[1], Jun Liu[1], Junsong Zhang[1], Xuanhong Zhang[1], Ryan Huang[1], Jiacong Zhao[1], Yonghong Li[2], Xuemei Ling[2], Yuewen Luo[1], Xiaoping Tang[2], Weiping Cai[2], Kai Deng[1], Linghua Li[2]*, Hui Zhang[1]*

[1]Institute of Human Virology, Key Laboratory of Tropical Disease Control of Ministry of Education, Guangdong Engineering Research Center for Antimicrobial Agent and Immunotechnology, Zhongshan School of Medicine, Sun Yat-sen University, Guangzhou, China; [2]Infectious Disease Center, Guangzhou Eighth People's Hospital, Guangzhou Medical University, Guangzhou, China

**Abstract** The antiviral activity of host factor apolipoprotein B mRNA editing enzyme catalytic polypeptide-like 3G (APOBEC3G, A3G) and its degradation mediated by human immunodeficiency virus type 1 (HIV-1) Vif protein are important topics. Although accumulating evidence indicates the importance of deubiquitination enzymes (DUBs) in innate immunity, it is unknown if they participate in A3G stability. Here, we found that USP49 directly interacts with A3G and efficiently removes ubiquitin, consequently increasing A3G protein expression and significantly enhancing its anti-HIV-1 activity. Unexpectedly, A3G degradation was also mediated by a Vif- and cullin-ring-independent pathway, which was effectively counteracted by USP49. Furthermore, clinical data suggested that USP49 is correlated with A3G protein expression and hypermutations in Vif-positive proviruses, and inversely with the intact provirus ratio in the HIV-1 latent reservoir. Our studies demonstrated a mechanism to effectively stabilize A3G expression, which could comprise a target to control HIV-1 infection and eradicate the latent reservoir.
DOI: https://doi.org/10.7554/eLife.48318.001

*For correspondence:
llheliza@126.com (LL);
zhangh92@mail.sysu.edu.cn (HZ)

**Competing interests:** The authors declare that no competing interests exist.

## Introduction

Human apolipoprotein-B-mRNA-editing enzyme catalytic polypeptide-like 3G (APOBEC3G, A3G) is a member of the cellular polynucleotide cytidine deaminase family. It can be incorporated into *vif*-deficient human immunodeficiency virus type1 (HIV-1) virions and mediates C–U conversion in the newly synthesized minus-stranded HIV-1 DNA to trigger the breakage of viral DNA or the generation of G-to A lethal hypermutations, resulting in a premature stop codon or mutated viral protein (*Harris et al., 2003*; *Mangeat et al., 2003*; *Sheehy et al., 2002*; *Zhang et al., 2003*). Alternatively, A3G can physically block the reverse transcription process (*Bishop et al., 2008*; *Iwatani et al., 2007*; *Pollpeter et al., 2018*). Because of these combined effects, A3G exerts potent antiviral activity. Meanwhile, it also induces sub-lethal hypermutations, which would not significantly impair viral infectivity but indicate an important driving force for HIV-1 genetic variations. This ultimately aids in viral evolution or facilitates viral responses to selective pressures such as the development of antiviral drug resistance or the evasion of immune surveillance (*Jern et al., 2009*; *Kim et al., 2010*; *Mulder et al., 2008*; *Sadler et al., 2010*; *Zhang et al., 2003*). However, the HIV-1 accessory protein viral infectivity factor (Vif) can effectively antagonize the antiviral activity of A3G by inducing its degradation through the ubiquitin–proteasome system (UPS) (*Kao et al., 2003*; *Mehle et al., 2004*;

*Sheehy et al., 2003*). Vif interacts with A3G through its N-terminal domain and has a SOCS-box motif within its C-terminal domain, which recruits ElonginB, ElonginC, and Cullin5 to form an E3 ubiquitin ligase complex, subsequently mediating the ubiquitination of A3G (*Yu et al., 2003*). CBF-β can increase the stability of HIV-1 Vif and promote assembly of the Vif-Cullin 5-E3-ubiquitin-ligase complex (*Jäger et al., 2011*; *Zhang et al., 2011*). Further, some clinical in vivo analyses of Vif-positive HIV-1 quasispecies indicated that A3G mediates lethal hypermutations and leads to the accumulation of G-to-A hypermutations in the Vif-positive HIV-1 genome, subsequently resulting in a large number of defective proviruses in the viral reservoir (*Borzooee et al., 2018*; *Cuevas et al., 2015*; *De Pasquale et al., 2013*; *Kieffer et al., 2005*). The expression level of A3G is also correlated with viremia and CD4 counts in the peripheral blood (*Amoêdo et al., 2011*; *Jin et al., 2005*; *Kourteva et al., 2012*; *Pace et al., 2006*; *Ulenga et al., 2008*; *Vázquez-Pérez et al., 2009*). Accordingly, partial mutations in the *vif* gene can impair its anti-A3G activity and increase the frequency of hypermutation (*Fourati et al., 2010*; *Simon et al., 2005*).

The homeostasis of eukaryotic cells is maintained by a well-tuned balance between the biosynthesis and degradation of proteins. Post-translational modification of proteins by ubiquitin (Ub) and their degradation by the UPS has emerged as a major regulatory process in virtually all aspects of cell biology. The UPS comprises the main non-lysosomal intracellular protein degradation pathway that consists of three major components, the proteasome holoenzymes, a number of ubiquitin ligases, and a broad spectrum of deubiquitinating enzymes (DUBs) (*Bailey-Elkin et al., 2017*; *Ventii and Wilkinson, 2008*). Ubiquitination is a dynamic and reversible process, and it has become increasingly apparent that deubiquitination also has an important function in regulating the ubiquitin-dependent pathway. However, In contrast to extensive studies on ubiquitination, the roles that deubiquitination play in host–virus relationships are less investigated (*Bailey-Elkin et al., 2017*; *Banks et al., 2003*; *Nijman et al., 2005*; *Sowa et al., 2009*). In addition to host DUBs, which play an important role in the maintenance of innate or acquired immunity, virus-encoded DUBs also affect the immune system to support their replication (*Kumari and Kumar, 2018*). Recently, some reports have indicated that DUBs directly bind viral proteins and affect their functions. For example, USP7 interacts with the Epstein–Barr nuclear antigen-1 (EBNA1) of Epstein-Barr virus and the ubiquitin E3 ligase ICP0 of the herpes simplex virus type-1 (HSV-1) to affect viral replication (*Berardi et al., 2004*; *Holowaty et al., 2003*). It also interacts with HIV-1 Tat and stabilizes this protein (*Ali et al., 2017*). Further, a USP14 inhibitor was found to inhibit the replication of Dengue viruses (*Nag and Finley, 2012*), whereas USP11 can interact with NP protein to inhibit influenza virus replication (*Liao et al., 2010*). However, the underlying mechanisms and their interactions with the USP have not been well elucidated.

Although the involvement of UPS in Vif-mediated A3G degradation is well known, whether DUB proteins participate in this important host-virus interaction is completely unknown. In this study, we screened a DUB-targeting siRNA library and found that USP49 participates in the stability of A3G protein. Further, we aimed to elucidate the association between USP49 and A3G in the presence of Vif, which could occur during natural HIV-1 infection. Our findings provide important insights into the role of USP49 in Vif-mediated A3G degradation. By enhancing the expression of A3G protein, USP49 exerts potent anti-HIV-1 activity and is involved in the generation of defective proviruses.

## Results

### Screening of a DUB-RNAi library identifies that USP18/41/49 regulates the expression of APOBEC3G

The human genome encodes approximately 100 putative DUBs. Based on the composition in relation to the catalytic motif, DUBs can be grouped into at least five subfamilies as follows: the ubiquitin-specific proteases/ubiquitin-specific processing proteases (USPs), the ubiquitin C-terminal hydrolases (UCHs), the ovarian tumor proteases (OTUs), the Josephin or Machado-Joseph disease protein domain proteases (MJDs), and the Jab1/MPN domain-associated metalloisopeptidase (JAMM) domain proteins (*Liu et al., 2018*; *Nijman et al., 2005*) (*Figure 1A*). We purchased a DUB siRNA library targeting 86 DUB genes and developed a cell-based high throughput system to screen DUB proteins that participate in the stability of an A3G-GFP fusion protein in 293 T cells (*Figure 1—figure supplement 1*). After two rounds of screening, we identified siRNAs specifically for USP18,

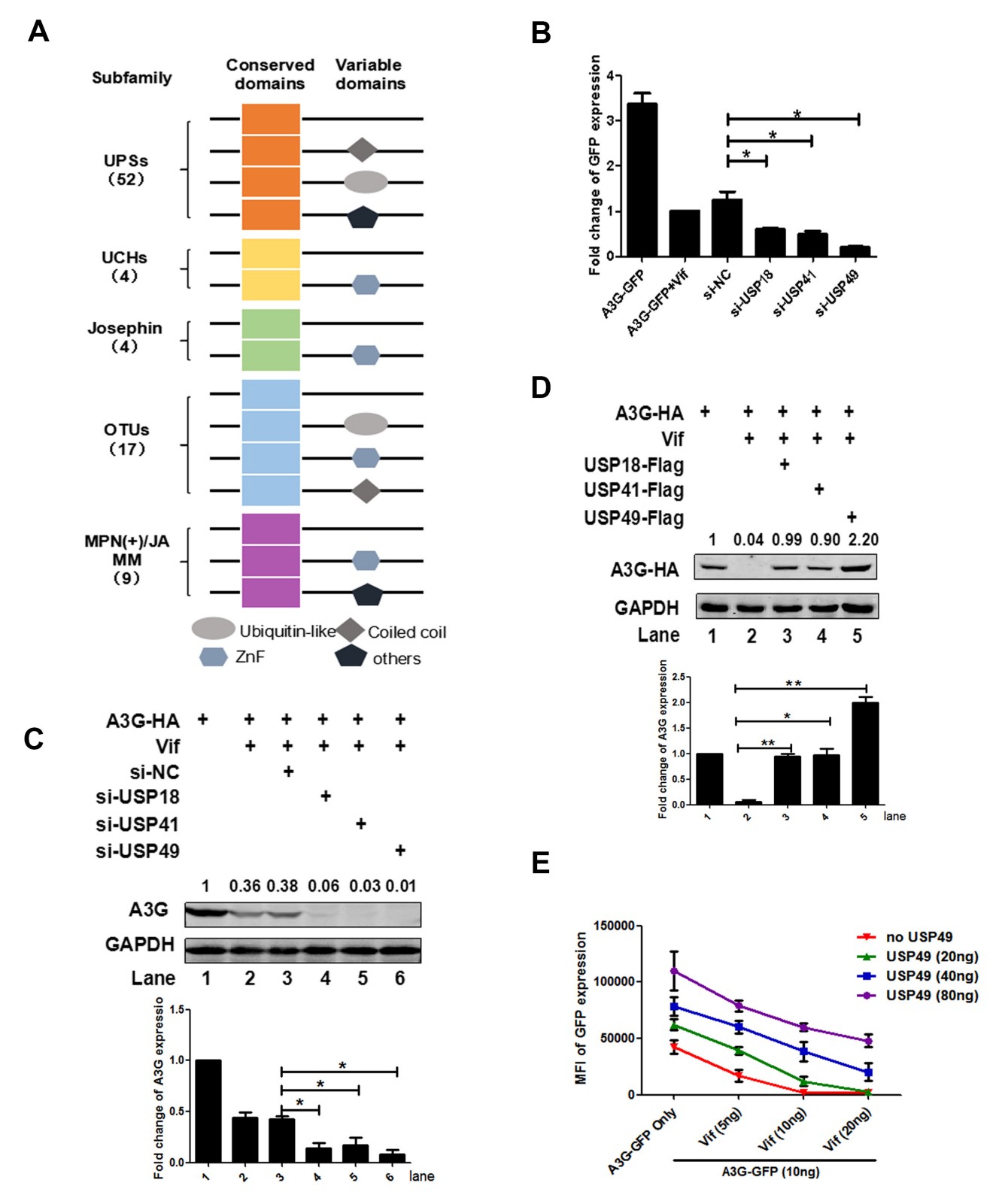

**Figure 1.** Screening of a DUB-RNAi library identifies that USP 18/41/49 regulates the expression of A3G. (**A**) Domain organization of five DUB subfamilies. (**B**) HEK293T cells were seeded into a 96-well plate with 20,000 cells/well, and then transfected with plasmids expressing A3G-GFP and Vif-HA, as well as siRNAs specific for USP18, USP41, or USP49 respectively. The GFP expression was detected with a PE Envision at 48 hr post-transfection. Error bars represent the SEM of three independent experiments.* p<0.05. (**C**) HEK293T cells were transfected with plasmids expressing A3G-HA and

*Figure 1 continued on next page*

*Figure 1 continued*

Vif-HA, as well as siRNAs specific for USP18, USP41, or USP49 respectively. After 48 hr, cells were lysed and Western blot was performed with the indicated antibodies. Representative data were shown and plotted with at least three independent experiments.* p<0.05. (**D**) HEK293T cells were transfected with plasmids expressing A3G-HA, Vif-HA, and one of plasmid expressing USP18-Flag, USP41-Flag, USP49-Flag. After 48 hr, cells were lysed and Western blot was performed with the indicated antibodies. Representative data were shown and plotted with at least three independent experiments.* p<0.05, **p<0.01. (**E**) HEK293T cells were transfected with indicated amounts of plasmids expressing A3G-GFP, Vif-HA, or USP49-Flag. The GFP expression was detected with a PE Envision at 48 hr post-transfection. Error bars represent the SEM of three independent experiments.

DOI: https://doi.org/10.7554/eLife.48318.002

The following figure supplements are available for figure 1:

**Figure supplement 1.** The schematic of high-throughput screening.
DOI: https://doi.org/10.7554/eLife.48318.003
**Figure supplement 2.** The distribution of GFP-tagged DUBs and A3G protein.
DOI: https://doi.org/10.7554/eLife.48318.004
**Figure supplement 3.** The expression of endogenous DUBs in primary CD4+ T cells.
DOI: https://doi.org/10.7554/eLife.48318.005
**Figure supplement 4.** The conservativeness analysis of DUBs among anthropoids.
DOI: https://doi.org/10.7554/eLife.48318.006

USP41, or USP49 that could facilitate Vif-mediated A3G degradation (*Figure 1B*).To further validate our screening results, we subsequently used DUB-specific siRNAs to knock down the expression of DUBs and co-transfected these with A3G-HA- and Vif-HA-expressing plasmids. After 48 hr, we performed a western blot to confirm that these DUBs could counteract Vif-mediated A3G degradation (*Figure 1C*). Further, the overexpression of USP18, USP41, or USP49 also supported this result. Among them, USP49 was the most capable of inhibiting this interaction (*Figure 1D*). Considering A3G exerts its antiviral effect in the cytoplasm (*Gallois-Montbrun et al., 2007*), we first constructed GFP-tagged DUB proteins. When these GFP-tagged proteins were transfected into HEK293T cells, we found that USP18 and USP41 were mainly distributed in the cytoplasm, whereas USP49 was distributed in both the nucleus and cytoplasm (*Figure 1—figure supplement 2*).

To define the biological relevance of these findings, we examined the expression of these DUBs in natural HIV-1-target cells and found that the expression of USP41 in primary CD4+T cells was quite low (*Figure 1—figure supplement 3A*), whereas the expression of USP18 and USP49 was high (*Figure 1—figure supplement 3A-C*). By comparing the sequences of these DUBs among anthropoids, we found that USP49 is the most conserved (*Figure 1—figure supplement 4A-B*). Thus, we chose USP49 as the major target for further investigations of mechanisms. To better understand the interaction between these DUBs and the Vif-A3G pathway, we co-transfected these three plasmids at different amounts into HEK293T cells. The results showed that USP49 significantly suppressed the downregulation of A3G when Vif levels were moderate. However, with an abundance of Vif protein, the protective effect of USP49 was quite limited (*Figure 1E*). Taken together, we identified that USP49 counteracts Vif-mediated A3G degradation in a dose-dependent manner.

## USP49 enhances the inhibitory effect of A3G on HIV-1 infectivity

To evaluate the inhibitory effect of DUB proteins on the Vif-positive HIV-1 replication, we co-transfected 293 T cells with USP49-specific siRNA, pNL4-3ΔVif, and various amounts of A3G- and Vif-expressing plasmids. After determining the dose-effect relationships among A3G, USP49, and Vif, shown in *Figure 1E*, we transfected Vif and A3G at a ratio 1:2. After normalization to HIV-1 p24 antigen for the resulting pseudotyped viruses, TZM-bl cells were infected. The results showed that the knockdown of endogenous USP49 could enhance the infectivity of HIV-1 in the presence of Vif (*Figure 2A*). When we transfected Vif and A3G at a ratio 1:1, the overexpression USP49 was found to inhibit HIV-1 infection and the inhibitory effect occurred in a dose-dependent manner (*Figure 2B*). Meanwhile, we sequenced the target motif of A3G on HIV-1 protease (the prot (nt 2280–2631)) region to detect hypermutations. The sequencing indicated that proviral DNAs in cells with a lower infectivity but a higher USP49 dose have more G-to-A hypermutations (*Figure 2—figure supplement 1A*) To explore the potential synergistic effect of USP49 and A3G on the infectivity of Vif-positive HIV-1 virions, HIV-1$_{NL4-3\Delta Env}$ particles were produced with increasing amounts of USP49 in the presence of A3G. After normalization to HIV-1 p24, TZM-bl cells were infected with

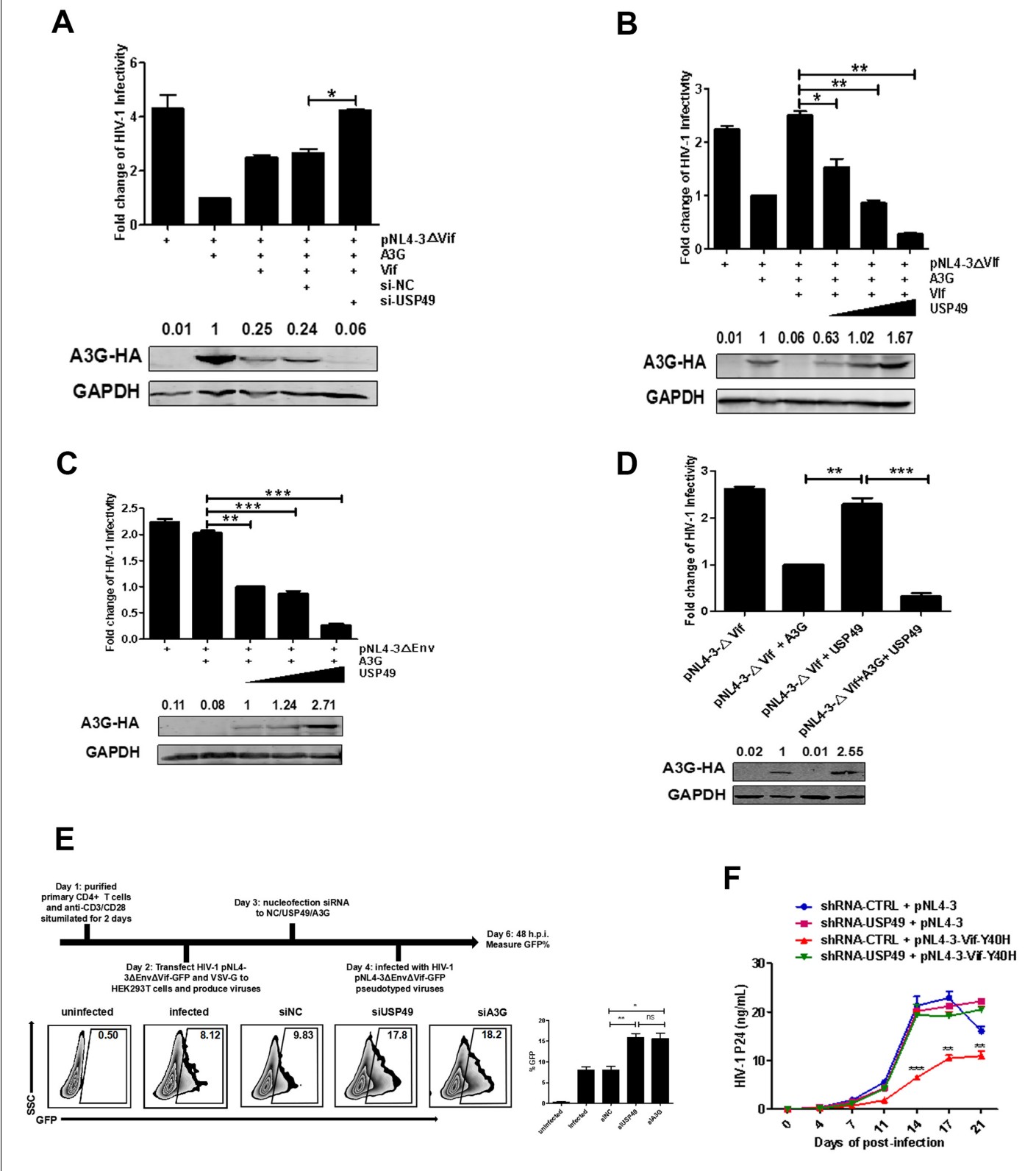

**Figure 2.** USP49 enhances the inhibitory effect of A3G on the infectivity of HIV-1. (**A**) HEK293T cells were first transfected with USP49-specific siRNA. After 12 hr, cells were co-transfected with pcDNA3.1-A3G-HA, pcDNA3.1-Vif-HA, and pNL4-3ΔVif. Culture supernatants were harvested at 72 hr post-transfection and then infected TZM-bl cells. After 48 hr, cells were harvested and the HIV-1 infectivity was detected by luciferase assay. Error bars represent the SEM of three independent experiments. *p<0.05. (**B**) HEK293T cells were co-transfected with pcDNA3.1-A3G-HA, pNL4-3ΔVif, and

*Figure 2 continued on next page*

*Figure 2 continued*

plasmids expressing Vif-HA or USP49-Flag respectively. Culture supernatants were harvested at 72 hr post-transfection and then allowed to infect TZM-bl cells. After 48 hr, cells were harvested and the HIV-1 infectivity was detected by luciferase assay. Error bars represent the SEM of three independent experiments. *p<0.05, **p<0.01. (C) HEK293T cells were co-transfected with pcDNA3.1-A3G-HA, pNL4-3ΔEnv, and different amounts of USP49-Flag-expressing plasmid respectively. Culture supernatants were harvested at 72 hr post-transfection and then infected TZM-bl cells. After 48 hr, cells were harvested and the HIV-1 infectivity was detected by luciferase assay. Error bars represent the SEM of three independent experiments. **p<0.01, ***p<0.001. (D) HEK293T cells were transfected with pNL4-3ΔVif, pcDNA3.1-A3G-HA plus pNL4-3ΔVif, or pNL4-3ΔVif plus USP49-Flag plasmids respectively. Culture supernatants were harvested at 72 hr post-transfection and then allowed to infect TZM-bl cells. After 48 hr, cells were harvested and the HIV-1 infectivity was detected by luciferase assay. Error bars represent the SEM of three independent experiments. ***p<0.001. (E) The primary CD4+T cells were stimulated with anti-CD3/28 for 2 days and then nucleofected with indicated siRNAs. After 24 hr, cells were infected with pNL4-3ΔEnvΔVif-GFP pseudotyped viruses. The infectivity were detected by flow cytometr on 48 h.p.i. Representative data were shown and plotted with at least three independent experiments.* p<0.05, **p<0.01. (F) HEK293T cells were transfected with pNL4-3 or Vif-Y40H-mutated-pNL4-3 respectively. Culture supernatants were harvested at 72 hr post-transfection and then allowed to infect shRNA-KD-USP49 primary CD4+ T cells. Cell supernatants were harvested for detecting HIV-1 P24 by ELISA Kit on several time points. Error bars represent the SEM of three independent experiments. The difference of p24 production from HIV-1$_{NL4-3VifY40H}$ infection between shRNA-NC and shRNA-KD-USP49 in primary CD4+ T cells at several time points was statistically analyzed. **p<0.01, ***p<0.001.

DOI: https://doi.org/10.7554/eLife.48318.007

The following figure supplements are available for figure 2:

**Figure supplement 1.** Analysis of A3G-induced hypermutation of *prot* DNA region.
DOI: https://doi.org/10.7554/eLife.48318.008

**Figure supplement 2.** The A3G protein level was detected by flow cytometry.
DOI: https://doi.org/10.7554/eLife.48318.009

these viral particles and virus infectivity was determined by luciferase assay. As shown in *Figure 2C*, USP49 regulated the infectivity of HIV-1 in a dose-dependent manner. Further, we generated viral particles overexpressing USP49 in the presence or absence of A3G. We found that USP49 alone did not have any antiviral effect (*Figure 2D*). Moreover, we examined the effects of USP49 on the viral infectivity with or without downregulation of endogenous A3G in primary CD4$^+$ T cells, which express both USP49 and endogenous A3G. We transfected siRNA with nucleofection to knockdown the endogenous USP49 or endogenous A3G level and then these cells were infected with HIV-1$_{NL4-3ΔEnvΔVif-GFP}$ pseudotyped viruses. Results showed that siUSP49 could down-regulate the endogenous A3G and then inhibit HIV-1 infectivity (*Figure 2E* and *Figure 2—figure supplement 2*).Combined with these data, this result indicates that USP49 exerts its potent antiviral effect through A3G.

Moreover, we generated an HIV-1 construct with the Vif-Y40H-mutant, which naturally occurs in HIV-1-infected patients and exhibits a weakened anti-A3G effect (*Simon et al., 2005*). When Vif-Y40H-mutated HIV-1 viruses were used to infect primary CD4+T cells, their infectivity was impaired (*Figure 2F*). However, in the primary CD4+T cells with shRNA-mediated USP49-knockdown, the infectivity of Vif-Y40H-mutated HIV-1 almost returned to wildtype levels (*Figure 2F*). Furthermore, we sequenced the proviral DNA in the cultures at the 21 day time point and analyzed the hypermutation with HYPERMUT soft online. We have not found any reversion of the Y40H mutation in the proviral DNA. Meanwhile, compared with suppression of USP49, there is a lower frequency of G-to-A hypermutation in proviral DNAs in the Y40H mutation control sample (*Figure 2—figure supplement 1B*). This result verifies that the inhibition of USP49 can promote the degradation of A3G and subsequently reduce the hypermutation in virus.These data indicated that the anti-A3G effect of Vif can be enforced by depletion of the A3G protein stabilizer USP49, alternatively supporting the fact that USP49 enhances the anti-HIV-1 effect of A3G.

## USP49 enhances the expression of A3G even in the absence of HIV-1 vif

Unexpectedly, in the absence of Vif, USP49 knockdown with siRNA also significantly decreased A3G stability, which could be rescued by MG132 (*Figure 3A,B*), suggesting that A3G could be degraded through a Vif-independent pathway. When DUBs were overexpressed, they promoted A3G protein expression in a dose-dependent manner (*Figure 3C*). Given that A3G can be packaged into HIV-1 virions to exert its anti-HIV-1 activity, we next evaluated the effects of USP49 on the levels of A3G in HIV-1 virions. For this, we co-transfected 293 T cells with A3G-HA- and USP49-Flag-expressing plasmids, along with pNL4-3ΔVif. USP49 enhanced A3G levels in both cell lysates and viral particles

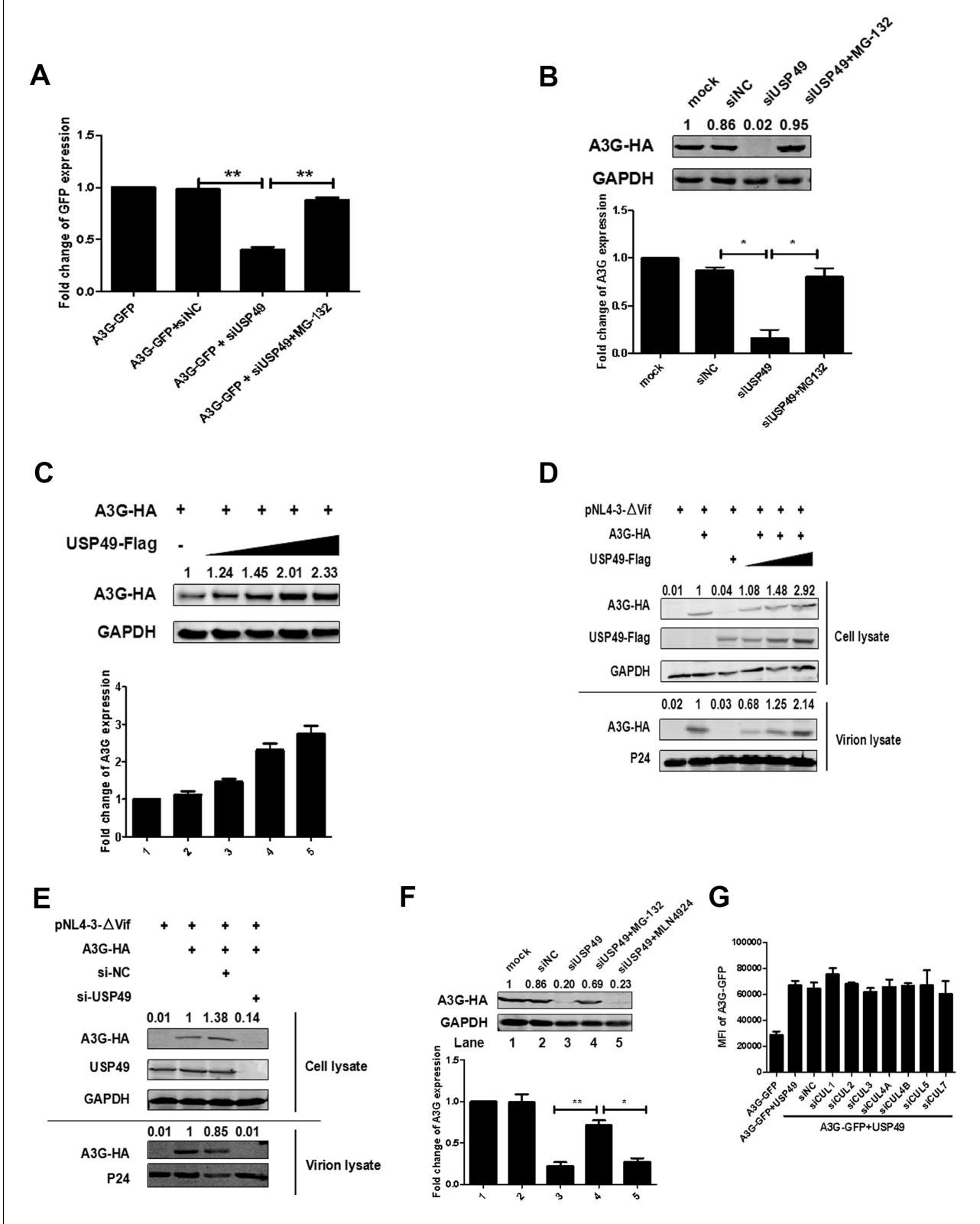

**Figure 3.** USP49 enhances the expression of A3G even in absence of Vif. (**A**) HEK293T cells were transfected with A3G-GFP-expressing plasmid and siRNA specific for USP49. MG132 was treated for 12 hr before harvest. The GFP expression was detected with a PE Envision at 48 hr post-transfection. Error bars represent the SEM of three independent experiments. **p<0.01. (**B**) HEK293T cells were transfected with A3G-HA-expressing plasmid and siRNA specific for USP49. MG132 was treated for 12 hr before harvest. After 48 hr post-transfection, cells were lysed and Western blot was performed

*Figure 3 continued on next page*

*Figure 3 continued*

with the indicated antibodies. Representative data were shown and plotted with at least three independent experiments.* p<0.05. (C) HEK293T cells were transfected with A3G-HA-expressing plasmid, and different amounts of USP49-Flag-expressing plasmid. After 48 hr, cells were lysed and Western blot was performed with the indicated antibodies. Representative data were shown and plotted with at least three independent experiments. (D) HEK293T cells were transfected with pcDNA3.1-A3G-HA, pcDNA3.1-USP49-Flag, and pNL4-3ΔVif. After 48 hr, cell pellets and supernatants were collected respectively. Cell pellets were lysed and subjected to immunoblotting with anti-HA, anti- Flag, and anti-GAPDH antibodies. Viral particles were collected from filtered supernatants by ultracentrifugation. The pelleted viral particles were lysed and detected by western blotting with anti-HA and anti-p24 antibodies. (E) HEK293T cells were transfected with pcDNA3.1-A3G-HA, pNL4-3, and a USP49-specific siRNA, After 48 hr, cell pellets and supernatants were collected respectively. Cell pellets were lysed and subjected to immunoblotting with anti-HA, anti- Flag, and anti-GAPDH antibodies. Viral particles were collected from filtered supernatants by ultracentrifugation. The pelleted viral particles were lysed and detected by western blotting with anti-HA and anti-p24 antibodies. (F) HEK293T cells were transfected with pcDNA3.1-A3G-HA and a USP49-specific siRNA. MG132 (10 uM) or MLN4924 (20 uM) was treated for 12 hr before harvest. After 48 hr post-transfection, cells were lysed and Western blot was performed with the indicated antibodies. Representative data were shown and plotted with at least three independent experiments.* p<0.05, **p<0.01. (G) HEK293T cells were transfected with pcDNA3.1-A3G-GFP, pcDNA3.1-USP49-FLAG and indicated siRNAs. The GFP expression was detected with a PE Envision at 48 hr post-transfection. Error bars represent the SEM of three independent experiments.

DOI: https://doi.org/10.7554/eLife.48318.010

The following figure supplements are available for figure 3:

**Figure supplement 1.** A3G can be packaged into HIV-1 virions with the help of USP49.

DOI: https://doi.org/10.7554/eLife.48318.011

**Figure supplement 2.** Detection of the knockdown efficiency of various siRNAs.

DOI: https://doi.org/10.7554/eLife.48318.012

(*Figure 3D*). Further, we have also performed a dose-dependent assay with over-expression of USP49 in the HIV-1$_{NL4-3}$ viruses and we have found an amount of A3G protein in the HIV-1$_{NL4-3}$ viral particles (*Figure 3—figure supplement 1*). Alternatively, the depletion of endogenous USP49 with USP49-specific siRNA also reduced the level of A3G in both ΔVif virus-producing cells and viral particles (*Figure 3E*). Taken together, we found that USP49 stabilizes A3G in both a Vif-dependent and Vif-independent manner.

Regarding the Vif-independent A3G degradation pathway, we speculated that an E3 protein and USP49 form a feedback loop to maintain the intracellular homeostasis of A3G under normal conditions. To determine the host E3 protein(s) that might be involved in A3G interactions, we first examined the possible involvement of the Vif-mediated Cullin5/Elongin B/C pathway. Although MG132 blocked this effect, a NEDD8 inhibitor MLN4924, which can inhibit the cullin-RING subtype of ubiquitin ligases, was ineffective (*Figure 3F*) (*Stanley et al., 2012*). Further, siRNAs specific for Cullin1–seven did not affect this process (*Figure 3G*). Moreover, we have also confirmed the knockdown efficiency of siRNA with qPCR (*Figure 3—figure supplement 2*). Collectively, although we cannot identify the specific E3 ligase at present, we at least excluded the involvement of the cullin-RING subtype of E3 ligases.

## USP49 directly interacts with A3G and deubiquitinates the K48-linked ubiquitination of A3G mediated by vif

Considering that USP49 could enhance the expression of A3G directly, we next assessed whether USP49 binds A3G. It has been reported that A3G is mainly distributed in the cytoplasm, where it exerts its antiviral effect; interestingly, USP49 was also found to be partly localized to the cytoplasm (*Figure 1—figure supplement 2*). Confocal laser scanning microscopy showed the colocalization of A3G and USP49 in the cytoplasm (*Figure 4A*), raising the possibility that USP49 and A3G interact directly. We therefore performed a co-immunoprecipitation experiment and found a strong interaction between Flag-USP49 and HA-A3G (*Figure 4B*). Meanwhile, HA-tagged A3G also interacted with endogenous USP49 (*Figure 4C*).We have also performed the experiment with RNase A treatment and found that the interaction between A3G and USP49 is direct and not mediated by RNA (*Figure 4D*). Since USP49 is a deubiquitination enzyme, we first examined whether it regulates the level of A3G ubiquitination in cells. As shown in *Figure 4E*, overexpression of USP49 resulted in a significant decrease in the poly-ubiquitination of A3G, whereas overexpression of the USP49-C262A mutant, which lost DUB enzymatic activity, failed to alter A3G ubiquitination (*Luo et al., 2017*). It is known that ubiquitin contains seven lysine (K) residues including K6, K11, K27, K29, K33, K48, and

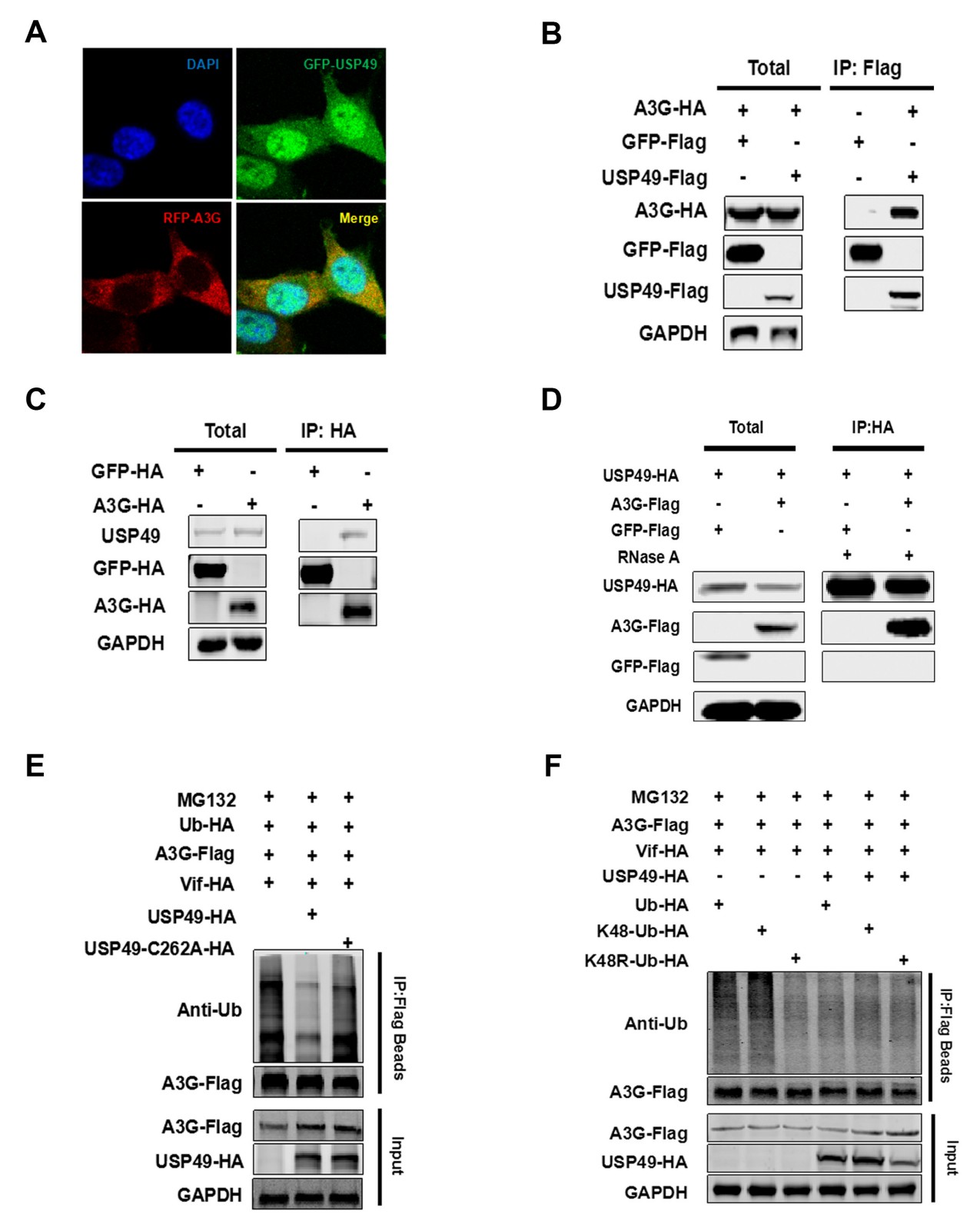

**Figure 4.** USP49 directly interacts with A3G and deubiquitinates the K48-linked ubiquitination of A3G. (**A**) HEK293T cells were co-transfected with plasmids expressing USP49-GFP or A3G-RFP. After 48 hr, the cells were fixed with 4% paraformaldehyde, and localization of USP49 and A3G was detected using confocal microscopy. (**B**) HEK293T cells were co-transfected with plasmids expressing Flag-USP49 or HA-A3G, or the indicated vectors. The cell lysates were immunoprecipitated using Flag beads and analyzed by immunoblotting with the indicated antibodies. (**C**) Hela cells were co-

*Figure 4 continued on next page*

*Figure 4 continued*

transfected with plasmids expressing HA-GFP or HA-A3G. The cell lysates were immunoprecipitated using HA beads and analyzed by immunoblotting with the indicated antibodies. (D) HEK293T cells were co-transfected with plasmids expressing HA-USP49 and Flag-A3G. The cell lysates were immunoprecipitated using HA beads and then treated with RNase A (20 ug/mL) for 1 hr. After that, samples were analyzed by immunoblotting with the indicated antibodies. (E) HEK293T cells were co-transfected with plasmids expressing USP49-HA, Vif-HA, Ub-HA, or A3G-Flag. Cells were treated with 10 uM MG132 for 24 hr before harvest. After 48 hr post-transfection, the cells were subjected to the denaturing immunoprecipitation using an anti-Flag beads followed by immunoblot analysis using the indicated antibodies. (F) HEK293T cells were transfected with the indicated plasmids expressing HA-ubiquitin, Flag-A3G, HA-Vif, or HA-USP49. After 24 hr, the cells were treated with 10 µM MG132 for 24 hr; the indicated types of ubiquitination were detected and quantified by the denaturing immunoprecipitation and western blotting.

DOI: https://doi.org/10.7554/eLife.48318.013

K63, through which poly-ubiquitin chains are linked to the substrate proteins. As the ubiquitination of A3G depends on the type of K48, we overexpressed K48 ubiquitin and confirmed that the ubiquitination of A3G is regulated by USP49. However, we found that the K48-linked ubiquitination was suppressed when the mutant K48R was used (*Figure 4F*), which is consistent with previous studies (*Mehle et al., 2004*; *Turner et al., 2016*).

## USP49 expression correlates with A3G and HIV-1 disease progression

The correlation between A3G or its hypermutation and important clinical parameters such as viral load and CD4 counts in the peripheral blood has been supported by multiple lines of evidence (*Amoêdo et al., 2011*; *Jin et al., 2005*; *Kourteva et al., 2012*; *Pace et al., 2006*; *Ulenga et al., 2008*; *Vázquez-Pérez et al., 2009*). This led us to investigate whether the expression of USP49 in host CD4+ T cells might affect the A3G-induced hypermutation in the HIV-1 genome by modulating the concentration of A3G protein. To determine the relevance of A3G regulation by USP49 in vivo, we performed intracellular staining to detect A3G protein levels in primary CD4+ T cells that were isolated from newly-diagnosed individuals with HIV-1 infections (n = 21). As shown in *Figure 5*, we observed a positive correlation between *USP49* mRNA levels and A3G protein expression (*Figure 5A*). Along with the correlation between A3G protein expression and CD4 counts, we also found a significant correlation between USP49 expression and CD4 counts (*Figure 5B,C*). Conversely, a negative correlation between plasma HIV-1 RNA levels and USP49 expression was observed (*Figure 5D*). However, we did not find any correlation between *USP18* mRNA expression and A3G, CD4 counts, or viral load (*Figure 5—figure supplement 1*). These results suggested that the expression of USP49 in CD4 T-cells of HIV-1 infected individuals could be involved in HIV-1 disease progression.

## USP49 affects the defective proviruses in HIV-1-latently-infected CD4 T-cells

Accumulating evidence indicates that A3G induces sub-lethal hypermutations in the Vif-positive HIV-1 genome and plays an important role in the production of defective proviral DNA in the HIV-1 reservoir (*Borzooee et al., 2018*; *Cuevas et al., 2015*; *De Pasquale et al., 2013*; *Kieffer et al., 2005*). A3G-induced hypermutations in *vif* and *env* genes of proviruses in vivo have been reported by several groups (*Fourati et al., 2010*; *Simon et al., 2005*). To study the influence of USP49 on hypermutation, the sequences of *vif* and *env*, amplified from the proviral DNA of CD4+T cells of seven HIV-1-infected individuals receiving suppressive cART, were used to determine the magnitude of A3G-mediated sublethal hypermutations in vivo. As a reference sequence, the virion-associated RNA from viral outgrowth of the same subject was also sequenced. The percentage of G-to-A mutations was quantified with an online Hypermut software. A positive correlation between the percentage of G-to-A mutations and USP49 expression and A3G protein expression was observed in these seven subjects (*Figure 6A–D*).

We next analyzed the expression of USP49 in CD4+T cells isolated from HIV-1-infected individuals receiving suppressive cART and its possible correlation with the ratio of defective proviruses in the resting CD4 T-lymphocytes. A newly-developed intact proviral DNA assay (IPDA) based on droplet digital PCR was used to quantitatively analyze intact proviruses and defective proviruses (n = 24; *Figure 6E*) (*Bruner et al., 2019*). By analyzing 24 clinical samples, we observed a significant negative correlation between *USP49* mRNA levels, as well as A3G protein expression, and the ratio of intact

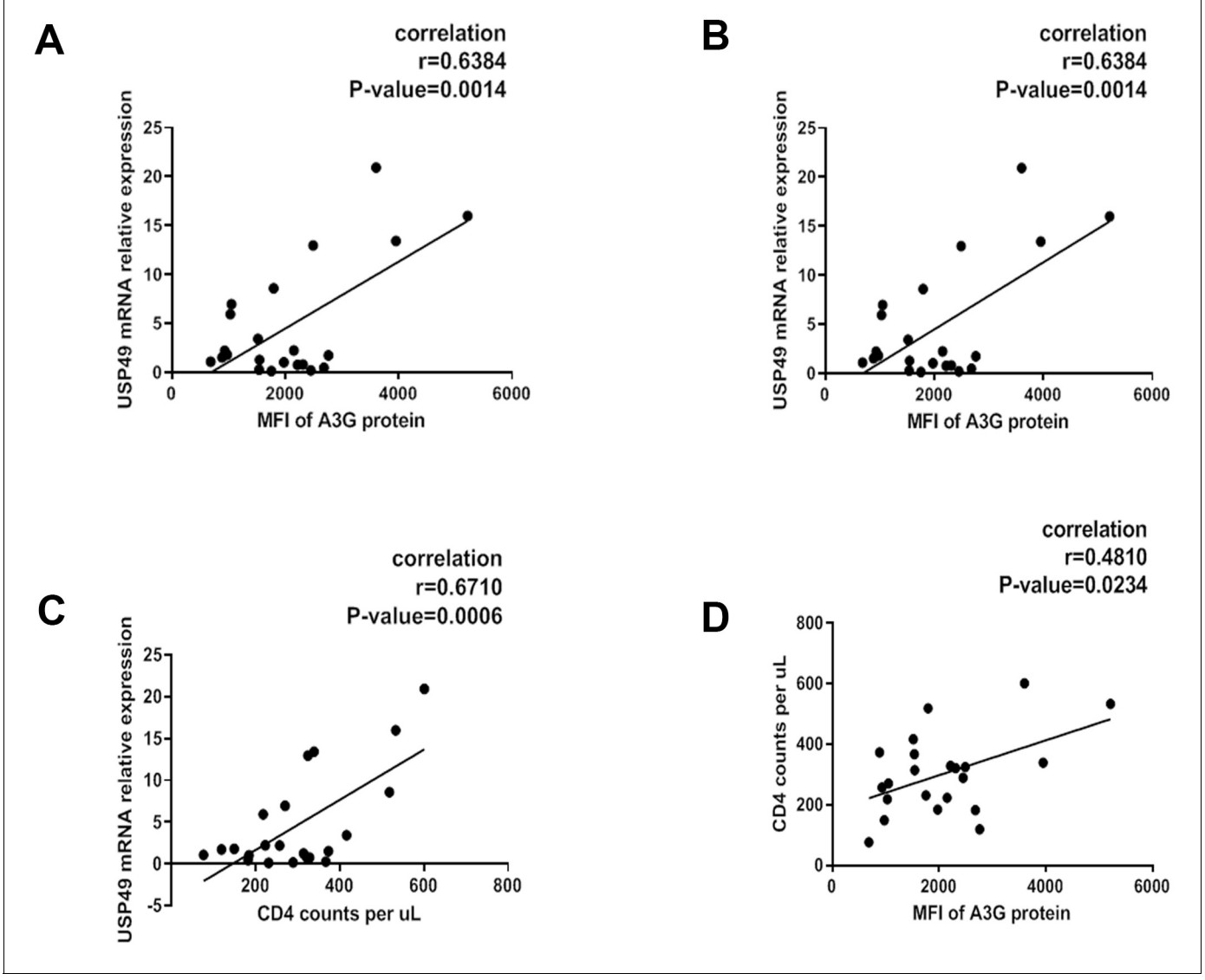

**Figure 5.** Associations of plasma HIV-1 RNA levels, CD4+ T cell counts and USP49 mRNA expression levels in the newly-diagnosed HIV-1-infected individuals. (A) Correlation between the expression of A3G protein level and the USP49 mRNA level in the CD4+T cells isolated from the newly-diagnosed HIV-1-infected individuals (n = 21). Pearson correlation coefficient and p value are listed. (B) Correlation between the count of CD4+T cells and the USP49 mRNA level in the CD4+T cells isolated from the newly-diagnosed HIV-1-infected individuals. Pearson correlation coefficient and p value are listed. (C) Correlation between the count of CD4+T cells and the A3G protein level in the CD4+T cells isolated from the newly-diagnosed HIV-1-infected individuals. Pearson correlation coefficient and p value are listed. (D) Correlation between the plasma HIV-1 RNA levels and the USP49 mRNA level in the CD4+T cells isolated from the newly-diagnosed HIV-1-infected individuals. Pearson correlation coefficient and p value are listed.
DOI: https://doi.org/10.7554/eLife.48318.014

The following figure supplement is available for figure 5:

**Figure supplement 1.** Associations of plasma HIV-1 RNA levels, CD4+ T cell counts, and USP18 mRNA expression levels in the newly-diagnosed HIV-1-infected individuals.
DOI: https://doi.org/10.7554/eLife.48318.015

proviruses (*Figure 6F,G*). Combined with the G to A hypermutation measurements, these data suggest that USP49 contributes to the generation of defective proviruses in the HIV-1 latently-infected cells and therefore the actual size of the viral reservoir.

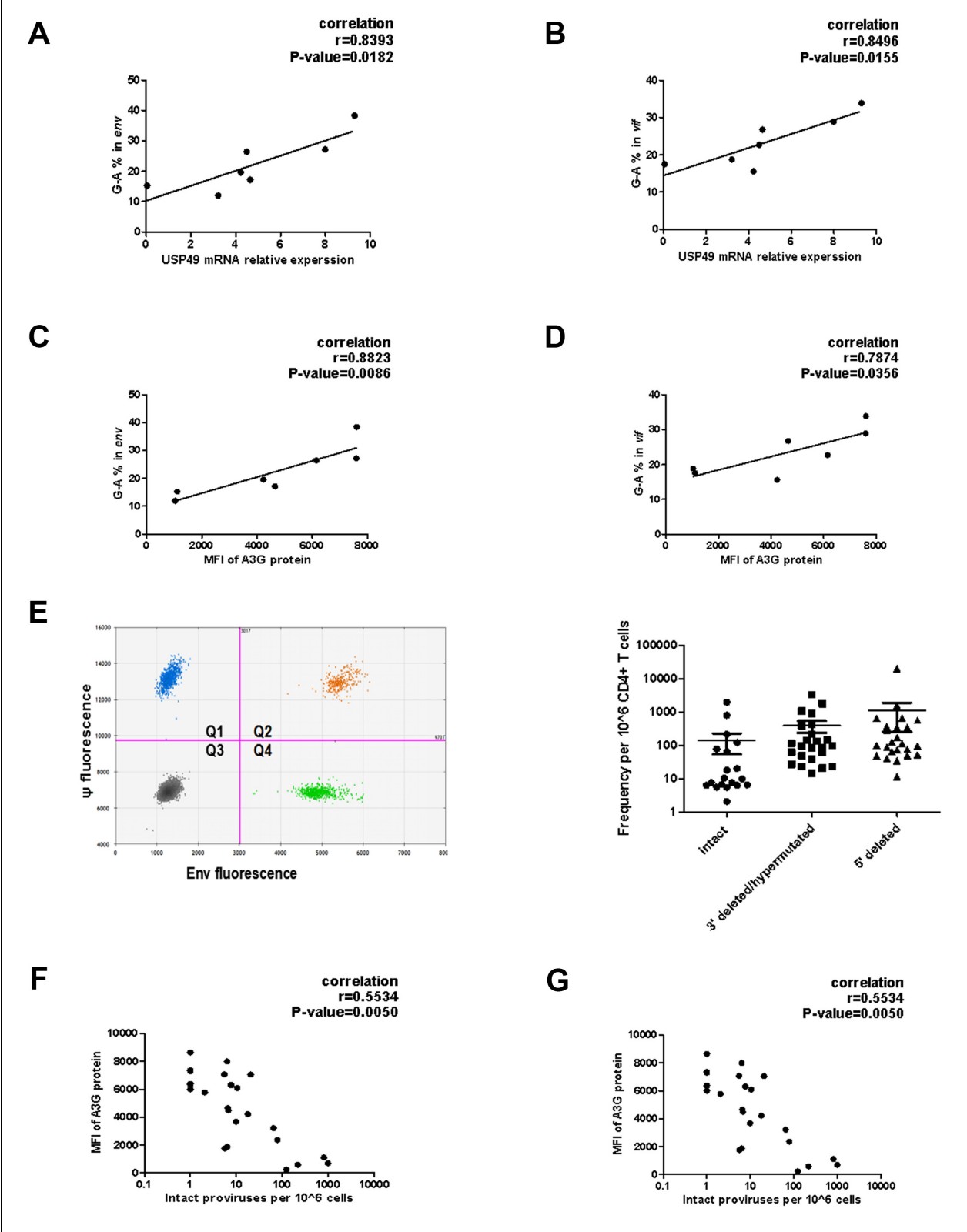

**Figure 6.** Associations of defective proviruses and USP49 mRNA expression levels in the CD4+ T cells isolated from HIV-1-infected individuals receiving suppressive cART. (A–B) The percentage of G-to-A in *env* and *vif* population sequences correlates with USP49 mRNA expression level in the CD4+ T cells from HIV-1-infected individuals (n = 7). Pearson correlation coefficient and p value are listed. (C–D) The percentage of G-to-A in *env* and *vif* population sequences correlates with A3G protein level in the CD4+ T cells from HIV-1-infected individuals (n = 7). Pearson correlation coefficient and p

*Figure 6 continued on next page*

*Figure 6 continued*

value are listed. (E) Representative IPDA results from a patient's CD4+ T cell sample. Boxed areas are expanded to show individual positive droplets (Left). IDPA results on CD4+ T cells from HIV-1-infected individuals (n = 24) with plasma HIV-1 RNA below the limit of detection (right). (F) Correlation between the intact proviruses and the A3G protein level in the CD4+T cells isolated from the indicated clinical HIV-1 patients. Pearson correlation coefficient and p value are listed. (G) Correlation between the intact proviruses and the USP49 mRNA level in the CD4+T cells isolated from the indicated clinical HIV-1 patients. Pearson correlation coefficient and p value are listed.

DOI: https://doi.org/10.7554/eLife.48318.016

## Discussion

USP49 is a DUB for which the function is relatively unknown. Recently, it has been shown that USP49 interacts with p53, dual-specificity protein phosphatases (DUSP), or FKBP51 in the nucleus, and consequently suppresses the ubiquitination of these proteins (*Luo et al., 2017*; *Tu et al., 2018*; *Zhang et al., 2019*). Regarding innate immunity, it can negatively regulate cellular antiviral responses via the deconjugation of K63-linked ubiquitin in STING, an adaptor protein that senses cytoplasmic DNA and is involved in various innate immune responses (*Ye et al., 2019*). Our data indicate that although USP49 is mainly distributed in the nucleus, it also exists in the cytoplasm. We also revealed a novel function for USP49 in the cytoplasm; specifically, it stabilizes A3G by counteracting Vif-mediated A3G ubiquitination by removing the ubiquitin from A3G. As a result, it significantly enhances the anti-HIV-1 activity of A3G. Our data also verified that its anti-HIV-1 effects are mainly due to the stabilization of A3G.

By studying the protection of A3G from degradation through DUB knockdown, we unexpectedly found that a Vif-independent degradation pathway for A3G. This pathway occurs via the UPS as MG132 could also block the degradation and ubiquitination of A3G. This phenomenon could explain why there are some ubiquitination sites on A3G that are not located at the C-terminus of A3G and not regulated by Vif (*Albin et al., 2013*; *Iwatani et al., 2009*; *Turner et al., 2016*). We attempted to identify the host E3 protein(s) that could be involved in A3G ubiquitination. At least, we found that the cullin family is not involved, as the inhibitor MLN4924 was ineffective and siRNAs for Cullin1-7 did not affect this process. Further, the depletion of elonginB and elonginC also did not affect the degradation of A3G, which clearly distinguishes this effect from the Vif-dependent degradation pathway.

It is well known that APOBEC proteins are not only involved in anti-HIV-1 immunity, but also counteract many viruses such as hepatitis B virus, adeno-associated virus, and some herpes viruses (*Bulliard et al., 2011*; *Nakaya et al., 2016*; *Narvaiza et al., 2009*; *Suspène et al., 2005*; *Xu et al., 2007*). They also play a key role in genome stability (*Esnault et al., 2005*; *Narvaiza et al., 2012*). Especially, they are involved in the development of mutations in tumors, which is a key factor for tumorigenesis and the development of drug-resistance or the escape from immune surveillance (*Burns et al., 2013a*; *Burns et al., 2013b*; *Law et al., 2016*; *Swanton et al., 2015*). It remains largely unknown how APOBEC3 proteins maintain their stability under these conditions. We identified that DUBs prevent the Vif-independent degradation of A3G, showing a new mechanism underlying the regulation of APOBE3 protein expression. Given the importance of the A3G protein and the potential functions of DUBs, future studies hold significant promise to exploit these novel mechanisms and develop therapeutic antiviral or antitumor agents. Nevertheless, considering that there are hundreds of E3 proteins, the identification of the E3 protein required for the Vif-independent degradation of A3G is interesting but beyond the scope of the current study.

The development of A3G-mediated sub-lethal hypermutations in the Vif-positive HIV-1 genome in vitro and in vivo has been widely accepted (*Borzooee et al., 2018*; *Cuevas et al., 2015*; *De Pasquale et al., 2013*; *Jern et al., 2009*; *Kieffer et al., 2005*; *Kim et al., 2010*; *Mulder et al., 2008*; *Sadler et al., 2010*; *Zhang et al., 2003*). Given that USP49 protects A3G from the Vif-dependent and Vif-independent degradation pathway, it is important to study the possible correlation between USP49 expression and A3G expression, and subsequently HIV-1 disease progression. In this report, we found a correlation between USP49 expression and A3G protein expression in vivo, as well as some important parameters related to HIV-1 disease progression and the components of the viral reservoir, through clinical correlation analysis. Although these data are preliminary, they support the hypothesis that the stabilization of A3G protein could lead to sub-lethal hypermutations

or eventually lethal hypermutations in the HIV-1 genome, resulting in the accumulation of defective proviruses. Therefore, USP49 and A3G could comprise a strong driving force to contain the actual viral reservoir. Moreover, our data indicated that the expression of USP49 protein in HIV-1-infected individuals could be used to indicate the quality of defective proviruses and the actual size of the viral reservoir, potentially serving as a possible new biomarker to evaluate the prognosis of HIV-1-infected patients. In summary, our work not only provides a new host antiviral mechanism, but also suggests an important opportunity for the development of new methods to control HIV-1 infection and eradicate the viral reservoir.

# Materials and methods

## Key resources table

| Reagent type (species) or resource | Designation | Source or reference | Identifiers | Additional information |
|---|---|---|---|---|
| Strain, strain background (*Escherichia coli*) | *E. coli* DH5α: F-, φ 80dlacZ ΔM15, Δ(lacZYA -argF) U169, deoR, recA1, endA1, hsdR17 (rK-, mK+), phoA, supE44, λ-, thi −1, gyrA96, relA1 | Takara | Cat#9057 | |
| Cell line (*Homo sapiens*) | HEK293T | ATCC | CRL-3216; RRID: CVCL_0063 | female |
| Cell line (*Homo sapiens*) | Hela | ATCC | CCL-2; RRID: CVCL_0030 | female |
| Cell line (*Homo sapiens*) | TZM-bl | NIH AIDS Reagent Program | Cat#8129 | female |
| Biological sample (*Homo sapiens*) | Blood samples from healthy individuals | Guangzhou Blood Center, Guangzhou | http://www.gzbc.org/ | |
| Biological sample (*Homo sapiens*) | Blood samples from HIV-1-infected individuals | Department of Infectious Diseases, Guangzhou 8th People's Hospital, Guangzhou | http://gz8h.com.cn/ | |
| Antibody | Mouse Monoclonal Anti-HA-Tag Antibody | MBL | Cat#M180-3 | Dilution 1:1000 |
| Antibody | Rabbit Anti-DDDDK Tag Polyclonal Antibody, Unconjugated | MBL | Cat#PM020 | Dilution 1:1000 |
| Antibody | Rabbit Polyclonal Anti-GAPDH Antibody | Proteintech | Cat#10494–1-AP | Dilution 1:1000 |
| Antibody | beta Actin Mouse McAb | Proteintech | Cat#66009–1-Ig | Dilution 1:1000 |
| Antibody | USP49 Rabbit Polyclonal antibody | Proteintech | Cat#18066–1-AP | Dilution 1:500 |
| Antibody | USP18 (D4E7) Rabbit mAb | Cell Signaling Technology (CST) | Cat#4813 | Dilution 1:1000 |
| Antibody | Anti-APOBEC3G/ A3G antibody | Abcam | Cat#ab75560 | Dilution 1:200 |
| Antibody | GFP (D5.1) XP Rabbit mAb | Cell Signaling Technology (CST) | Cat#2956 | Dilution 1:1000 |
| Antibody | Ubiquitin Rabbit Polyclonal antibody | Proteintech | Cat#10201–2-AP | Dilution 1:1000 |
| Antibody | IRDye 680RD Goat anti-Mouse IgG (H + L), 0.5 mg Antibody | LI-COR Biosciences | Cat#926–68070 | Dilution 1:10000 |

*Continued on next page*

*Continued*

| Reagent type (species) or resource | Designation | Source or reference | Identifiers | Additional information |
|---|---|---|---|---|
| Antibody | IRDye 800CW Goat Anti-Rabbit IgG, Conjugated Antibody | LI-COR Biosciences | Cat#926–32211 | Dilution 1:10000 |
| Antibody | Goat Anti-Mouse IgG H and L (DyLight 488) preadsorbed | Abcam | Cat#ab96879 | Dilution 1:500 |
| Antibody | Goat Anti-Mouse IgG H and L (DyLight 594) preadsorbed | Abcam | Cat#ab96881 | Dilution 1:500 |
| Antibody | EZviewTM Red Anti-HA Affinity Gel | Sigma-Aldrich | Cat# A2220-10ML | 30 ul/sample |
| Antibody | ANTI-FLAG M2 Affinity Gel | Sigma-Aldrich | Cat# E6779-1ML | 30 ul/sample |
| Recombinant DNA reagent | VSV-G glycoprotein-expression vector | PMID: 9306402 | Addgene Plasmid #12259 | Dr. Didier Trono (School of Life Sciences, Ecole Polytechnique Fédérale de Lausanne, Lausanne, Switzerland) |
| Recombinant DNA reagent | Lentiviral packaging construct pCMVΔR8.2 | PMID: 9306402 | Addgene Plasmid #12263 | Dr. Didier Trono (School of Life Sciences, Ecole Polytechnique Fédérale de Lausanne, Lausanne, Switzerland) |
| Sequence-based reagent | siRNA Library | RiboBio | http://www.ribobio.com/ | |
| Chemical compound, drug | TRIzolTM Reagent | ThermoFisher | Cat#15596018 | |
| Chemical compound, drug | Triton X-100 | Sigma-Aldrich | Cat#T8787-50ML | |
| Chemical compound, drug | Penicillin-Streptomycin,Liquid | ThermoFisher | Cat#15140122 | |
| Commercial assay or kit | BD IMag Human CD4+ T Lymphocyte Enrichment Set-DM | BD Biosciences | Cat#557939 | |
| Commercial assay or kit | HIV-1 p24 ELISA Kit | Abcam | Cat#ab218268 | |
| Software, algorithm | Prism 5 | GraphPad | https://www.graphpad.com/scientific-software/prism/ | |
| Software, algorithm | FlowJo V10 | Tree Star | https://www.flowjo.com/ | |
| Software, algorithm | Odyssey CLX Imager | LI-COR Biosciences | https://www.licor.com/bio/products/imaging_systems/odyssey/ | |
| Software, algorithm | Image Studio Lite Ver 4.0 | LI-COR Biosciences | https://www.licor.com/bio/products/software/image_studio_lite/ | |
| Software, algorithm | Hypermut | | https://www.hiv.lanl.gov/content/sequence/HYPERMUT/hypermut.html | |

## Plasmid constructions

HIS-FLAG-USP49 was purchased from Vigene bioscience (cat# CH806995). USP49C262A mutant was generated by PCR-based site-directed mutagenesis. Other USP49 deletion mutants were sub-cloned into pcDNA3.1 vector with HA or FLAG tag. The vector pLKO.3G, which contains a U6 promoter and *gfp* selection gene was purchase from Addgene, was used for expression of USP49 shRNA or scrambled control. The pGFP-USP49 was generated by sub-cloning *usp49* gene into pEGFP-C1

vector. The pcDNA3.1-A3G-HA/FLAG, pcDNA3.1-GFP-HA/FLAG, and pcDNA3.1-ub-HA/FLAG, pet28a-Vif and pet32a-A3G were constructed as described previously (*Chen et al., 2017*). The pNL4-3-Vif-Y40H mutant was generated by digestion of pNL4-3 with Apa I and EcoR I, and followed by PCR-based site-directed mutagenesis. All the constructs were confirmed by sequencing.

## Cell culture and transfection

HEK293T and Hela cells were obtained from ATCC. TZM-bl cells were obtained from NIH AIDS Reagent Program.. All cells have been tested for mycoplasma using a PCR assay and confirmed to be mycoplasma-free. These cells were grown at 37°C with 5% CO2 in Dulbecco's modified Eagle's medium (DMEM) (Invitrogen) supplemented with 10% fetal bovine serum (FBS) (Invitrogen) and 1% penicillin–streptomycin (Invitrogen). The cells were transfected with the indicated plasmids or siRNAs by lipofectamine 2000 (Invitrogen). The procedures described by the manufacturer were followed.

## DUB siRNA library screening

All the smart pools of siRNAs were obtained from RiboBio (Guangzhou, China). These siRNAs were utilized for screening. Briefly, HEK293T cells are seeded into a 96-well plate with 20,000 cells/well, and then transfected with plasmids A3G-GFP, Vif-HA and various siRNAs per well. The GFP expression was detected with a PE Envision (PerkineElmer) at 48 hr post-transfection.

## Co–immunoprecipitation and western blotting

Co-IP and western blotting assays were performed as previously described (*Pan et al., 2018*). In brief, cells were lysed with the lysis buffer (150 mM NaCl, 50 mM Tris–HCl [pH 7.5], 1 mM EDTA,1% Triton X-100, 0.5% NP-40), plus PMSF and protease inhibitor cocktail for 30 min at 4°C. The cell lysates were clarified by centrifugation at 18,000 g for 30 min at 4°C, then mixed with anti-HA or anti-Flag agarose beads (Sigma) and incubated at 4°C for 4 hr, followed by washing four times with cold lysis buffer and eluting in gel loading buffer. The immunoprecipitated samples were analyzed by SDS-PAGE and detected by western blotting. Quantity One program (Biorad) was used to quantify the western blotting results. The information of antibodies were shown in Key resources table in the supplemental material.

## The purification of pseudotyped HIV–one viruses

Human 293 T cells were transfected with pNL4-3-ΔEnvGFP or pNL4-3-ΔVif and other indicated plasmids. After 48 hr of transfection, cell supernatants were collected, centrifuged at 4°C for 10 min at 8000 rpm (≈ 7000 g) and filtered through a 0.45 μm filter to remove cellular debris. Then the cell-free supernatants were concentrated by ultracentrifugation through a 20% sucrose cushion at 4°C for 2 hr at 45,000 rpm (≈ 40,000 g) (HITACHI Preparative Ultracentrifuge, CP80WX). The pellets were re-suspended in RIPA buffer containing protease inhibitor cocktail and subjected to immunoblotting.

## Virus infectivity assay

Human 293 T cells were co-transfected with pNL4-3-ΔVif or pVSV-G plus pNL4-3ΔEnv-GFP, in the presence or absence of pcDNA3.1-A3G-HA or pcDNA3.1-USP49-FLAG. The virus-containing supernatant were collected at 48 hr after transfection and filtered by a 0.45 μm filter. After normalization for HIV-1 p24 by enzyme-linked immunosorbent assay (ELISA, Clonetech), TZM-bl cells ($2.5 \times 10^5$ cells per well in 24-well plates) were infected with viruses which containing 5 ng of p24 antigen. And then, luciferase enzyme activity was determined at 72 hr post infection.

## In vivo deubiquitination assay

For the in vivo deubiquitination assay, HEK293T cells were co-transfected with plasmids expressing HA-USP49 wild-type (WT) or HA-USP49 with a mutant at Cys 262 to Ala (C262A mutant), Flag-A3G, and Ub-HA. After 2 days, cells were treated with MG132 (10 uM) for 24 hr before being harvested. The cell extracts were subjected to immunoprecipitation with the indicated antibodies for 4 hr at 4°C. Then, the precipitated immunocomplexes were separated by SDS–PAGE and blotted with anti-Ub and anti-A3G antibody.

## Human subjects

Peripheral blood for the isolation of primary CD4 cells was obtained from HIV-1-infected individuals. One part of HIV-1-infected individuals had been on cART for at least 12 months and had maintained undetectable HIV-1 viremia (<50 HIV-1 RNA copies per ml of plasma). Another part was the newly diagnosed HIV-1 patients with high HIV-1 RNA copies in plasma. All the HIV-1-infected individuals were recruited from Guangzhou Eighth People's Hospital. Buffy coats derived from the blood of healthy donors were used in in vitro experiments. All human samples were anonymously coded in accordance with the local ethical guidelines (as stipulated by the Declaration of Helsinki). The Ethics Review Board of Sun Yat-Sen University and the Ethics Review Board of Guangzhou 8th People's Hospital approved this study. Written informed consents were provided by all study participants, and the protocol was approved by the IRB of Guangzhou Eighth People's Hospital (Guangzhou, China).

## Isolation and culture of primary CD4+ T cell and HIV-1 infection

Peripheral blood mononuclear cells (PBMCs) were obtained from healthy donors and isolated using Ficoll gradient centrifugation, followed by culturing in the conditioned RPMI 1640 medium. The CD4 + T cells were then isolated by MACS microbead-negative sorting using human CD4+ T cell isolation kit (BD Bioscience). The purity of CD4+ T cell fraction was higher than 95%. Then the CD4+ T cells from healthy donors were activated by 1 µg/ml anti-CD3/CD28 mAbs and 100 U/ml IL-2 for 3 days, followed by infection with HIV-1$_{NL4-3}$, HIV-1$_{NL4-3VifY40H}$, or HIV-1 reporter viruses. The HIV-1 viruses were generated by transfecting 293 T cells with pNL4-3 or pNL4-3 VifY40H. Productive infection was determined by P24 ELISA kit.

## RNA isolation and real-time PCR (qRT-PCR)

Cellular viral DNA/RNA was isolated from the CD4 T-cells of HIV-1 infected individuals according to the instructions using AllPure DNA/RNA Micro Kit (Magen, R5112-02). Then cDNA was synthesized using a SuperScript III reverse transcriptase kit (Qiagen, Valencia, CA). The qRT-PCR reactions were performed in triplicate using SYBR Green (TaKaRa, Otsu, Japan) and normalized to endogenous actin and GAPDH mRNA levels using gene-specific primers for each target (see *Supplementary file 1*).

## Nested RT-PCR and sequencing

RNA from CD4+ T cells was extracted with an RNase minikit (Qiagen) and used for amplification of viral *vif* or partial *env* genes (V1-V3 region). The first-round PCR was performed using a one-step RT-PCR kit (TaKaRa, Otsu, Japan), and second-round PCR was performed with High Fidelity Prime Star (TaKaRa, Otsu, Japan), following the manufacturer's instructions. The primers used for this experiment are listed in *Supplementary file 1*. For clonal sequencing, PCR products were TA cloned into the pMD-18 T vector (TaKaRa, Otsu, Japan) and 10 clones were sequenced for each sample.

## Hypermutation analysis

The percentages of G-to-A mutations were quantified with online Hypermut software (www.hiv.lanl.gov/content/sequence/HYPERMUT/hypermut.html) (*Rose and Korber, 2000*; *Simon et al., 2005*). Virion-associated RNA from viral outgrowth of the same subject was also sequenced as a reference sequence.

## Intact proviral DNA assay (IPDA)

The procedures for IPDA described previously were followed with minor modifications (*Bruner et al., 2019*). In general, the IPDA is performed on DNA from $2 \times 10^6$ CD4+ T cells. Genomic DNA is extracted using the QIAamp DNA Mini Kit (Qiagen) with precautions to avoid excess DNA fragmentation. Quantification of intact, 5'deleted, and 3'deleted and/or hypermutated proviruses was carried out using primer/probe combinations optimized for subtype B HIV-1. The primer/probe mix consists of oligonucleotides for two independent hydrolysis probe reactions that interrogate conserved regions of the HIV-1 genome to discriminate intact from defective proviruses (*Supplementary file 1*). HIV-1 reaction A targets the packaging signal (Ψ) that is a frequent site of small deletions and is included in many large deletions in the proviral genome. The Ψ amplicon is

positioned at HXB2 coordinates 692–797. This reaction uses forward and reverse primers, as well as a 5'6-FAM-labeled hydrolysis probe. Successful amplification of HIV-1 reaction A produced FAM fluorescence in droplets containing Ψ, detectable in channel 1 of the droplet reader. HIV-1 reaction B targets the RRE of the proviral genome, with the amplicon positioned at HXB2 coordinates 7736–7851. This reaction used forward and reverse primers, as well as two hydrolysis probes: a 5'VIC-labeled probe specific for wild-type proviral sequences, and a 5'unlabelled probe specific for APO-BEC3G hypermutated proviral sequences (*Supplementary file 1*). Successful amplification of HIV-1 reaction B produced a VIC fluorescence in droplets containing a wild-type form of RRE, detectable in channel 2 of the droplet reader, whereas droplets containing a hypermutated form of RRE were not fluorescent.

Droplets containing HIV-1 proviruses were scored as follows. Droplets positive for FAM fluorescence only, which arises from Ψ amplification, was scored as containing 3' defective proviruses, with the defect attributable to either APOBEC3G mediated hypermutation or 3' deletion. Droplets positive for VIC fluorescence only, which arises from wild-type RRE amplification, was scored as containing 5' defective proviruses, with the defect attributable to 5' deletion. Droplets positive for both FAM and VIC fluorescence was scored as containing intact proviruses. Double-negative droplets contained no proviruses or rare proviruses with defects affecting both amplicons.

## Digital droplet PCR

The procedures for ddPCR described previously were followed with minor modifications (*Bruner et al., 2019*). Briefly, The ddPCR was performed on the Bio-Rad QX200 AutoDG Digital Droplet PCR system using the appropriate manufacturer supplied consumables and the ddPCR Supermix for probes (no dUTPS) (Bio-Rad Laboratories). For HIV-1 proviral discrimination reactions, 600 ng of genomic DNA was analyzed in each reaction well. Uninfected donor CD4+T cells was performed for each IPDA run as a negative control while qualified genomic DNA of J-Lat 6.3 cell line was analyzed in each IPDA run as a positive control. Multiple replicate wells were performed for each reaction type to ensure a consistent quantification, and replicate wells were merged during analysis to increase IPDA dynamic range. Results are expressed as intact proviral DNA copies per $10^6$ CD4+ T cells.

## Statistics

Statistics analysis were performed with GraphPad Prism 6. All of data were reported as mean ± SEM. For multiple comparisons, a one-way followed by the Bonferroni's correction (only two groups were compared) was applied. Correlation was estimated by Pearson correlation coefficients (for parametric data). Differences were found to be significant when P was less than 0.05, 0.01, or 0.001, as indicated by single (*), double (**), or triple asterisks (***) within the figures. Most graphs were produced using Prism.

## Acknowledgements

This work was supported by the National Special Research Program of China for Important Infectious Diseases (2018Z × 10302103 and 2017Z × 10202102), the Important Key Program of the Natural Science Foundation of China (81730060), the International Collaboration Program of the Natural Science Foundation of China and the US NIH (81561128007), and the Joint-Innovation Program in Healthcare for Special Scientific Research Projects of Guangzhou (201803040002) to HZ. This work was also supported by the National Natural Science Foundation of China (8197080527), the Science and Technology Planning Project of Guangzhou (201704020226), and the Pearl River S and T Nova Program of Guangzhou (201806010118) to TP.

## Additional information

### Funding

| Funder | Grant reference number | Author |
|---|---|---|
| National Special Research Program of China for Important Infectious Diseases | 2018ZX10302103 | Hui Zhang |
| National Special Research Program of China for Important Infectious Diseases | 2017ZX10202102 | Hui Zhang |
| National Natural Science Foundation of China | Key Program 81730060 | Hui Zhang |
| The International Collaboration Program of the Natural Science Foundation of China and the US NIH | 81561128007 | Hui Zhang |
| The Joint-Innovation Program in Healthcare for Special Scientific Research Projects of Guangzhou | 201803040002 | Hui Zhang |
| The Science and Technology Planning Project of Guangzhou | 201704020226 | Ting Pan |
| Pearl River S and T Nova Program of Guangzhou | 201806010118 | Ting Pan |
| National Natural Science Foundation of China | 81971918 | Ting Pan |

The funders had no role in study design, data collection and interpretation, or the decision to submit the work for publication.

### Author contributions

Ting Pan, Data curation, Formal analysis, Funding acquisition, Validation, Investigation, Methodology, Writing—original draft, Project administration; Zheng Song, Resources, Data curation, Formal analysis, Validation, Methodology; Liyang Wu, Data curation, Formal analysis, Validation; Guangyan Liu, Resources, Data curation, Formal analysis, Validation; Xiancai Ma, Zhilin Peng, Yonghong Li, Resources, Validation, Methodology; Mo Zhou, Bingfeng Liu, Resources, Data curation, Validation; Liting Liang, Data curation, Validation, Methodology; Jun Liu, Resources, Software; Junsong Zhang, Resources, Software, Methodology; Xuanhong Zhang, Jiacong Zhao, Yuewen Luo, Kai Deng, Resources, Methodology; Ryan Huang, Investigation, Writing—review and editing; Xuemei Ling, Xiaoping Tang, Weiping Cai, Linghua Li, Resources, Investigation, Methodology; Hui Zhang, Funding acquisition, Investigation, Project administration, Writing—review and editing

### Author ORCIDs

Ting Pan (iD) https://orcid.org/0000-0002-7106-7312
Guangyan Liu (iD) https://orcid.org/0000-0002-5891-1830
Xiancai Ma (iD) http://orcid.org/0000-0002-4934-4221
Hui Zhang (iD) https://orcid.org/0000-0003-3620-610X

### Ethics

Human subjects: All human samples were anonymously coded in accordance with the local ethical guidelines (as stipulated by the Declaration of Helsinki). The Ethics Review Board of Sun Yat-Sen University and the Ethics Review Board of Guangzhou 8th People's Hospital approved this study.Written informed consents were provided by all study participants, and the protocol was approved by the IRB of Guangzhou Eighth People's Hospital (Guangzhou, China).

Decision letter and Author response
Decision letter https://doi.org/10.7554/eLife.48318.020
Author response https://doi.org/10.7554/eLife.48318.021

## Additional files

### Supplementary files

• Supplementary file 1. The sequences of primers and probes.
DOI: https://doi.org/10.7554/eLife.48318.017

• Transparent reporting form
DOI: https://doi.org/10.7554/eLife.48318.018

### Data availability

All data generated or analysed during this study are included in the manuscript and supporting files.

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
