## [Decision Letter]

Thank you for submitting your article "USP49 potently stabilizes APOBEC3G protein and inhibits HIV-1 replication" for consideration by *eLife*. Your article has been reviewed by two peer reviewers, one of whom is a member of our Board of Reviewing Editors, and the evaluation has been overseen by Päivi Ojala as the Senior Editor. The reviewers have opted to remain anonymous.

The reviewers have discussed the reviews with one another and the Reviewing Editor has drafted this decision to help you prepare a revised submission.

Summary:

In this paper, the authors identified the deubiquitinase USP49 as a factor that stabilizes the host antiviral factor APOBEC3G (A3G). USP49 interacted with and removes ubiquitins from A3G, and thereby prevents the degradation of A3G. Analysis of clinical samples of HIV-1 infected patients revealed that USP49 expression is directly correlated with hypermutation rate and inversely correlated with intact proviruses in latent reservoirs. While the reviewers found the paper interesting, some essential revisions are needed before it can be published.

Essential revisions:

1) The authors should provide more solid and systematic data to show that USP49 inhibits HIV-1 infectivity through A3G. Particularly, in Figure 2B, please show by Western blotting that A3G levels are rescued in cells and its virion incorporation is increased in a USP49 dose-dependent manner. Furthermore, please show that the inhibition of viral replication is associated with increased hypermutation of the proviral DNAs in target cells.

2) The authors should provide more quantitative data to show that the effects are significant. Specifically, the authors should quantitate the Western blot bands and plot the results of 2-3 independent experiments. Representative Western blots with adequate exposure may be needed such that the effects may look more obvious. This requirement applies to Figure 1C,D, and Figure 3B,E,F. New experiments may not be needed here.

3) The authors should show that USP49 works with the endogenous A3G to inhibit HIV-1 infectivity. Particularly, in Figure 2D, the authors should compare effects of USP49 on the viral infectivity with or without downregualtion of endogenous A3G. A different cell line may be needed in which the endogenous A3G is expressed at a functionally detectable level.

The full reviews are listed below. The rationale for the revisions may be better understood in the context of the full reviews. In addition, the comments should be helpful for the authors to improve the manuscript.

*Reviewer #1:*

In this paper, the authors screened siRNAs targeting deubiquitinases for those involved in Vif-mediated degradation of A3G and identified USP49 as one that promoted the stability of A3G. The authors further demonstrated that USP49 interacted with A3G and reduced its ubiquitination. In HIV-infected patients, the expression levels of USP49 correlated well with A3G expression and disease progression parameters including CD4^+^ T cell counts and viral loads. Furthermore, the expression levels of USP49 correlated with the hypermutation rates of the Vif-positive proviruses. These are important findings that would be interesting to a broad audience of *eLife*, though the paper should be better written and some concerns need to be addressed before publication.

1) Figure 2D. USP49 is expected to aid the endogenous A3G to inhibit HIV-1 infectivity. But we do not see that effect here. One explanation is that in these cells there is little or no endogenous A3G. In this case a cell line that expresses A3G at a level functionally detectable should be used to analyze the effect of USP49 with or without A3G downregulation. In the whole paper, A3G was overexpressed. It is important to show that USP49 works with the endogenous A3G to inhibit HIV infectivity.

2) Figure 3E. Was USP49-Flag co-expressed in this experiment? If no, where is the USP49-Flag panel from? If yes, it should be indicated in the figure legend. In the latter case, the experiment would have been too complicated. Maybe the endogenous ESP49 was mistakenly labeled as USP49-Flag?

*Reviewer #2:*

Deubiquitinating enzymes (DUBs) can regulate the stability of proteins. The authors found that a DUB, USP49, directly interacts with A3G and removes ubiquitins from A3G, increasing its steady state levels and its antiviral activity. It was also observed that Vif can mediate degradation of A3G by a cullin-ring-independent pathway that is counteracted by USP49. Analysis of clinical samples revealed that USP49 expression is directly correlated with hypermutation and inversely correlated with intact proviruses in latent reservoirs.

Overall, some innovative experiments that explore the role of deubiquitinating enzymes in Vif-mediated degradation of APOBEC3G are described. While some of the results are strong, others are not as convincing. Additional results that support and strengthen some major conclusions are needed.

Figure 1E: At the low dose of Vif, GFP expression is increased to ~1.7-fold compared to the no Vif control. Is this due to a Vif-independent effect or is this just variation between experiments? Three independent transfection experiments should be done and provide error bars, and to establish the reproducibility of the results.

It is stated that USP49 significantly suppressed downregulation of A3G at moderate levels of Vif, but the effect was limited at high levels of Vif. However, the fold increase in the GFP expression seems higher when high levels of Vif were transfected.

Figure 2B: The dose-dependent suppression of infectivity with increasing levels of USP49 in the presence of A3G and Vif is convincing. The authors should provide western blots in parallel to show a dose-dependent increase in steady-state levels of A3G in cells and in virion-incorporated A3G. In addition, the authors should provide data indicating that the dose-dependent decrease in infectivity is accompanied with higher frequencies of G-to-A hypermutation of proviral DNAs in infected cells.

Figure 2E: A Y40H Vif mutant was used to carry out the experiments over 21 days. The authors should sequence the proviral DNA in the cultures at the 21-day time point to verify that the rescue in virus was not due to reversion of the Y40H mutation and due to suppression of USP49.

Figures 3D and 3E: the rescue in A3G-HA levels in cells and in virion lysates is very mild and not convincing. As stated above, a dose-dependent rescue of A3G levels and their virion incorporation should be shown by performing quantitative western blots and at least three independent experiments.

Figure 3F: The result that MLN4924 did not rescue A3G levels is not convincing; multiple experiments with quantitative western blots should be shown.

Figure 3G: Did the authors verify that the siRNA knockdowns of the cullins were effective in reducing the steady-state levels of cullins? How efficient was the knockdown and is it possible that residual amounts of cullins remaining were sufficient to induce degradation of A3G-GFP?

The vertical axis should indicate that a fold decrease in GFP expression was being measured to avoid confusion.

Figure 4B and 4C: The IP experiments should be performed with and without RNase treatment to show that the interaction between A3G and USP49 is direct and not an indirect interaction mediated by both proteins binding to RNA.

Figure 6E – 6G: More informative figure legends and methods description are needed to clarify these results.

---

## [Author Response]

Essential revisions:1) The authors should provide more solid and systematic data to show that USP49 inhibits HIV-1 infectivity through A3G. Particularly, in Figure 2B, please show by Western blotting that A3G levels are rescued in cells and its virion incorporation is increased in a USP49 dose-dependent manner. Furthermore, please show that the inhibition of viral replication is associated with increased hypermutation of the proviral DNAs in target cells.

We appreciate the suggestions and apologize for missing the solid Western blot data in our results which we have now corrected. We would like to emphasize our revisions by the following points:

A) In the new Figure 2A-2C, we illustrated that A3G levels in the cell were rescued by USP49, through Western blotting. Meanwhile, we have also quantitated the Western blot bands and retrieved data from three independent experiments.

B) In the newly-added Figure 2—figure supplement 1A, we sequenced the target motif of A3G on HIV-1 protease (the *prot* (nt 2280-2631)) region to detect hypermutations. Proviral DNAs from samples with reduced infectivity have more G-to-A hypermutations, which are correlated with A3G protein level and USP49 level.

2) The authors should provide more quantitative data to show that the effects are significant. Specifically, the authors should quantitate the Western blot bands and plot the results of 2-3 independent experiments. Representative Western blots with adequate exposure may be needed such that the effects may look more obvious. This requirement applies to Figure 1C,D, and Figure 3B,E,F. New experiments may not be needed here.

According to the editors and reviewer’s suggestions, all of these experiments (including Figure 1C,D and Figure 3B,C,F) shown here now represent at least three independent experiments (results with adequate exposure) and we quantitated the Western blot bands with Image J software.

3) The authors should show that USP49 works with the endogenous A3G to inhibit HIV-1 infectivity. Particularly, in Figure 2D, the authors should compare effects of USP49 on the viral infectivity with or without downregualtion of endogenous A3G. A different cell line may be needed in which the endogenous A3G is expressed at a functionally detectable level.

Thanks for this suggestion. We originally used 293T cell line even though it does not have endogenous A3G. Although H9 cell line serves as a typical non-permissive cell and is frequently used for endogenous A3G study, it has a low expression of USP49 protein level (data not shown). Per suggestion, we decided to examine the effects of USP49 on the viral infectivity with or without downregulation of endogenous A3G in primary CD4^+^ T cells, which express both endogenous USP49 and endogenous A3G. We transfected siRNA with nucleofection to knockdown the endogenous USP49 or endogenous A3G level and then these cells were infected with HIV-1_NL4-3ΔEnv_ pseudotyped viruses. Results in the new Figure 2E showed that USP49 worked with endogenous A3G to inhibit HIV-1 infectivity. Furthermore, the intracellular staining indicated the expression of A3G protein levels were decreased by the siRNA knockdown of USP49.

Reviewer #1:

In this paper, the authors screened siRNAs targeting deubiquitinases for those involved in Vif-mediated degradation of A3G and identified USP49 as one that promoted the stability of A3G. The authors further demonstrated that USP49 interacted with A3G and reduced its ubiquitination. In HIV-infected patients, the expression levels of USP49 correlated well with A3G expression and disease progression parameters including CD4^+^ T cell counts and viral loads. Furthermore, the expression levels of USP49 correlated with the hypermutation rates of the Vif-positive proviruses. These are important findings that would be interesting to a broad audience of eLife, though the paper should be better written and some concerns need to be addressed before publication.

Thanks for these positive comments. We have gone through the whole manuscript very carefully to eliminate any grammar or presentational errors.

1) Figure 2D. USP49 is expected to aid the endogenous A3G to inhibit HIV-1 infectivity. But we do not see that effect here. One explanation is that in these cells there is little or no endogenous A3G. In this case a cell line that expresses A3G at a level functionally detectable should be used to analyze the effect of USP49 with or without A3G downregulation. In the whole paper, A3G was overexpressed. It is important to show that USP49 works with the endogenous A3G to inhibit HIV infectivity.

Thanks for the reviewer’s comments. We originally used 293T cell line even though it does not have endogenous A3G. Although H9 cell line serves as a typical non-permissive cell and is frequently used for endogenous A3G study, it has a low expression of USP49 protein level (data not shown). Per suggestion, we decided to examine the effects of USP49 on the viral infectivity with or without downregulation of endogenous A3G in primary CD4^+^ T cells, which express both endogenous USP49 and endogenous A3G. We transfected siRNA with nucleofection to knockdown the endogenous USP49 or endogenous A3G level and then these cells were infected with HIV-1_NL4-3ΔEnv_ pseudotyped viruses. Results in the new Figure 2E showed that USP49 worked with endogenous A3G to inhibit HIV-1 infectivity. Considering the nucleofection and infection may lead to many cell death and the cells is insufficient for western blot, we performed intracellular staining of A3G protein. The flow cytometry data indicated the expression of A3G protein levels were decreased by the siRNA knockdown of USP49 and A3G (Figure 2—figure supplement 2).

2) Figure 3E. Was USP49-Flag co-expressed in this experiment? If no, where is the USP49-Flag panel from? If yes, it should be indicated in the figure legend. In the latter case, the experiment would have been too complicated. Maybe the endogenous ESP49 was mistakenly labeled as USP49-Flag?

We apologize for this mistake. The endogenous USP49 was mistakenly labeled as USP49-Flag. We have carefully corrected all the presentational mistakes in the revised manuscript. We will remember this lesson and will be extremely careful in all our research works and post-work presentations.

Reviewer #2:

Deubiquitinating enzymes (DUBs) can regulate the stability of proteins. The authors found that a DUB, USP49, directly interacts with A3G and removes ubiquitins from A3G, increasing its steady state levels and its antiviral activity. It was also observed that Vif can mediate degradation of A3G by a cullin-ring-independent pathway that is counteracted by USP49. Analysis of clinical samples revealed that USP49 expression is directly correlated with hypermutation and inversely correlated with intact proviruses in latent reservoirs.Overall, some innovative experiments that explore the role of deubiquitinating enzymes in Vif-mediated degradation of APOBEC3G are described. While some of the results are strong, others are not as convincing. Additional results that support and strengthen some major conclusions are needed.

Thanks for the constructive comments. We have performed additional experiments to provide more solid data and support our major conclusions.

Figure 1E: At the low dose of Vif, GFP expression is increased to ~1.7-fold compared to the no Vif control. Is this due to a Vif-independent effect or is this just variation between experiments? Three independent transfection experiments should be done and provide error bars, and to establish the reproducibility of the results.

To address the reviewer’s concerns, we have repeated this experiment many times with proper amounts of plasmids. Through many times of experiments, we feel that the approximate 1.7-fold increase observed in the previous work was due to the variations between the experiments. The Vif-independent effect is observed at the 1^st^ lane, which indicates that while the amount of USP49 increased, A3G level also increased in the absence of Vif expression.

The new Figure 1E were representative of three independent experiments and the error bars were provided. Meanwhile, we have corrected the Y axis title as “MFI of A3G-GFP expression” in the revised Figure for better understanding.

It is stated that USP49 significantly suppressed downregulation of A3G at moderate levels of Vif, but the effect was limited at high levels of Vif. However, the fold increase in the GFP expression seems higher when high levels of Vif were transfected.

Thanks for the comments and we have adjusted the amount of three plasmids (emphasize contrast in the levels) and obtained a more significant result in the new Figure 1E.

Figure 2B: The dose-dependent suppression of infectivity with increasing levels of USP49 in the presence of A3G and Vif is convincing. The authors should provide western blots in parallel to show a dose-dependent increase in steady-state levels of A3G in cells and in virion-incorporated A3G. In addition, the authors should provide data indicating that the dose-dependent decrease in infectivity is accompanied with higher frequencies of G-to-A hypermutation of proviral DNAs in infected cells.

Thanks for the suggestions and we have now added the western blotting data in our results. In the new Figure 2B, we have shown through Western blot that A3G levels were rescued in cell by USP49 in dose-dependent manner. Meanwhile, we have also quantitated the Western blot bands and the results were representative of three independent experiments. Meanwhile, in the new Figure 2—figure supplement 1A, we sequenced the target motif of A3G on HIV-1 protease (the *prot* (nt 2280-2631)) region to detect hypermutations. The sequencing indicated that proviral DNAs in cells with lower infectivity (has a higher USP49 dose) have more G-to-A hypermutations. These G-to-A hypermutations were caused by A3G protein, which was correlated with USP49 level.

Figure 2E: A Y40H Vif mutant was used to carry out the experiments over 21 days. The authors should sequence the proviral DNA in the cultures at the 21-day time point to verify that the rescue in virus was not due to reversion of the Y40H mutation and due to suppression of USP49.

Thanks for the reviewer’s suggestion, we sequenced the proviral DNA in the cultures at the 21-day time point and analyzed the hypermutation. The results are shown in the new Figure 2—figure supplement 1B. We have not found any reversion of the Y40H mutation in the cell-free virion RNA. Meanwhile, compared with suppression of USP49, there is a lower frequency of G-to-A hypermutation in proviral DNAs in the Y40H mutation control sample. This result verifies that the inhibition of USP49 can promote the degradation of A3G and subsequently reduce the hypermutation in virus.

Figures 3D and 3E: the rescue in A3G-HA levels in cells and in virion lysates is very mild and not convincing. As stated above, a dose-dependent rescue of A3G levels and their virion incorporation should be shown by performing quantitative western blots and at least three independent experiments.

Thanks for the reviewer’s suggestion, we have repeated the experiments regarding Figure 3D and 3E. Meanwhile, according to the reviewer’s suggestion, we have performed a dose-dependent assay with over-expression of USP49 in the new Figure 3D. All these experiments shown here represent at least three independent experiments and we quantitated the Western blot bands of A3G with Image J.

Figure 3F: The result that MLN4924 did not rescue A3G levels is not convincing; multiple experiments with quantitative western blots should be shown.

According to the reviewer’s suggestion, we have repeated the western blot multiple times and quantitated the Western blot bands with Image J. All these experiments shown here represent at least three independent experiments and representative data were showed in the new Figure 3F. The result now clearly indicates that MLN4924 did not rescue A3G levels.

Figure 3G: Did the authors verify that the siRNA knockdowns of the cullins were effective in reducing the steady-state levels of cullins? How efficient was the knockdown and is it possible that residual amounts of cullins remaining were sufficient to induce degradation of A3G-GFP?

Thanks for the reviewer’s comment. To address your concerns, we first detected the knockdown efficiency of these siRNAs by qRT-PCR and the results demonstrated that the knockdown efficiency of siRNA is more than 70% (Figure 3—figure supplement 2A).

Moreover, two more experiments proved the efficiency of siRNAs (Figure 3—figure supplement 2B,C). One was that the siRNA knockdown of Cullin5 decreased the degradation of A3G through Vif-dependent way. This indicates that the siRNA for Cullin5 was very effective. Another experiment was that the expression of AID increased when Cullin7 was inhibited by the siRNA. Cullin7 is involved in the degradation of AID, which was reported by our group recently (Luo et al., 2019). In both experiments, the effects of Cullins were significantly inhibited by siRNA. If these Cullins are involved in the Vif-independent degradation of A3G, the current 70% knockdown efficiency would be enough to cause detectable changes in the results. However, we did not see any effect of the Cullin1-7 knockdown (by siRNA) on the A3G-GFP expression. Therefore, according to our existing experimental results, we believe that Cullin1-7 do not participate in the Vif-independent degradation pathway of A3G.

The vertical axis should indicate that a fold decrease in GFP expression was being measured to avoid confusion.

Thanks for the reviewer’s suggestion. We have corrected the vertical axis to “MFI of GFP Expression” in the revised Figure 3G.

Figure 4B and 4C: The IP experiments should be performed with and without RNase treatment to show that the interaction between A3G and USP49 is direct and not an indirect interaction mediated by both proteins binding to RNA.

Thanks for the reviewer’s suggestion. We have performed the experiment with RNase A treatment in the new Figure 4D and found that the interaction between A3G and USP49 is direct and not mediated by RNA.

Figure 6E – 6G: More informative figure legends and methods description are needed to clarify these results.

Thanks for the reviewer’s comments. We have described the details of IPDA methods in the revised manuscript. These methods were reported by Siliciano’s lab in 2019 (Katherine et al., 2019). It is a quantitative approach for measuring the reservoir of latent HIV-1 proviruses.

In general, the IPDA was performed on DNA from 2 × 10^6^ CD4^+^ T cells. Genomic DNA was extracted using the QIAamp DNA Mini Kit (Qiagen) with precautions to avoid excess DNA fragmentation. Quantification of intact, 5’deleted, and 3’deleted and/or hypermutated proviruses was carried out using primer/probe combinations optimized for subtype B HIV-1. The primer/probe mix consists of oligonucleotides for two independent hydrolysis probe reactions that interrogate conserved regions of the HIV-1 genome to discriminate intact from defective proviruses (Supplementary file 2). HIV-1 reaction A targets the packaging signal (Ψ) that is a frequent site of small deletions and is included in many large deletions in the proviral genome. The Ψ amplicon is positioned at HXB2 coordinates 692–797. This reaction used forward and reverse primers, as well as a 5’6-FAM-labelled hydrolysis probe. Successful amplification of HIV-1 reaction A produced FAM fluorescence in droplets containing Ψ, detectable in channel 1 of the droplet reader. HIV-1 reaction B targets the RRE of the proviral genome, with the amplicon positioned at HXB2 coordinates 7736-7851. This reaction used forward and reverse primers, as well as two hydrolysis probes: a 5’VIC-labelled probe specific for wild-type proviral sequences, and a 5’unlabelled probe specific for APOBEC3G hypermutated proviral sequences (Supplementary file 2). Successful amplification of HIV-1 reaction B produced a VIC fluorescence in droplets containing a wild-type form of RRE, detectable in channel 2 of the droplet reader, whereas droplets containing a hypermutated form of RRE were not fluorescent.

Droplets containing HIV-1 proviruses were scored as follows. Droplets positive for FAM fluorescence only, which arises from Ψ amplification, was scored as containing 3′ defective proviruses, with the defect attributable to either APOBEC3G mediated hypermutation or 3′ deletion. Droplets positive for VIC fluorescence only, which arises from wild-type RRE amplification, was scored as containing 5′ defective proviruses, with the defect attributable to 5′ deletion. Droplets positive for both FAM and VIC fluorescence was scored as containing intact proviruses. Double-negative droplets contained no proviruses or rare proviruses with defects affecting both amplicons.

These explanations were added to the manuscript in the Materials and methods section.